# Comparative Sensitivity Analysis of Hydrology and Relative Corn Yield under Different Subsurface Drainage Design Using DRAINMOD

Haribansha Timalsina, Soonho Hwang *, Richard A. Cooke and Rabin Bhattarai

Department of Agricultural and Biological Engineering, University of Illinois at Urbana-Champaign, 1304 W Pennsylvania Ave, Urbana, IL 61801, USA; ht15@illinois.edu (H.T.); rcooke@illinois.edu (R.A.C.); rbhatta2@illinois.edu (R.B.)
* Correspondence: soonho@illinois.edu

**Abstract:** DRAINMOD is a process-based hydrologic model used to analyze the effectiveness of various drainage systems and management strategies. In this study, a sensitivity analysis of DRAINMOD hydrologic parameters for two different field settings located at Champaign, Illinois, was performed to determine the most sensitive parameters that affect the subsurface flow and relative productivity of corn. Latin-Hypercube One-Factor-at-a-Time (LH-OAT) was used to determine the sensitivity index of 17 parameters for six objective functions for daily flow, water balance, and relative yield for the productivity of corn. The results indicated that flow and yield were highly sensitive to drainage design parameters such as drainage depth and spacing. Winter flow and the water balance were sensitive to soil thermal conductivity parameters; however, they had no impact on the relative corn yield. The significant difference in sensitivity of the two fields was observed in the hydraulic conductivity of soil layers due to varying thicknesses for different soil types. This study highlights the need for more careful calibration of these sensitive parameters to reduce equifinality and model output uncertainty and appropriate drainage design for optimizing crop productivity and drainage outflow.

**Keywords:** DRAINMOD; uncertainty; LH-OAT sensitivity; equifinality; multi-objective calibration

## 1. Introduction

For facilitating crop production and improved yield in areas with inadequate agricultural drainage, subsurface (tile) drainage helps assure reliable and successful crop production by removing extra water from the agricultural field. By enhancing trafficability, drainage makes it easier to access fields in time for activities like tillage, planting, and harvesting. Plant stress can be also reduced by removing the extra water from the root zone and boosting crop output [1]. In addition, subsurface drainage lessens surface runoff, sediment losses, and the flow of contaminants adhering to the sediment, including phosphorus, nitrogen, and pesticides, into surface waters [2]. It enhances soil aeration at the root depth, which assists in organic matter decomposition in the soil.

Due to the interconnected dynamics of soil, water, and plant systems with multiple processes and variables that influence the behavior, monitoring studies on a large scale to capture this complexity can be challenging. Hydrological models have proliferated in the past few decades as a tool to evaluate the problems associated with water management primarily for two reasons. First, using models enhances the current understanding of cause-effect dynamics in the aquatic ecosystem. Second, models offer a synthesis of essential insights in the policy arena [3]. The process-based hydrological models conceptualize the scientific understanding of the watershed's and field's hydrological, plant physiological, and biogeochemical processes, providing an edge over empirical models for simulating these associated processes [4].

No hydrological models are an ideal representation of the actual process involved, and it is also challenging to provide the initial and boundary conditions required by a model with absolute precision. Thus, all the model calibrations and future predictions/projections are subjected to uncertainty which arises primarily due to input parameters, observed data, and uncertainties in the structure of model [5]. On the other hand, different sets of model parameters lead to a similar performance index (for example, NSE or RSQ), which leads to equifinality [6]. The bias in model predictions and the likelihood of errors increases with the increase in uncertainty of the input parameters.

Due to the incompatibility between the model complexity and observed data, determining the model input parameters during the calibration phase is one of the critical tasks regardless of the hydrological model used. For example, the model parameters in hydrological models can be estimated using manual and autocalibration approaches using discharge-related measures. However, if the hydrological model has lots of parameters, calibrating numerous parameters by manual is a time-consuming and cumbersome process. Parameter sensitivity analysis can be considered to effectively minimize adjustment workload and achieve optimum simulation quickly and efficiently direct field data collection and monitoring [7,8].

Sensitivity analysis is widely used to quantify the impact of change in model parameters in the variance in predicting model outputs. It provides insights into the parameters that have the most bearing on the model outputs, which can be utilized to refine calibration and improve the model structure to lower model complexity and uncertainty [4]. Primarily, two methods, global sensitivity and local sensitivity, are applied to study the sensitivity of the models. In global sensitivity, all parameters are simultaneously altered in each model run. In contrast to global methods, in the local sensitivity analysis (also called the one-at-a-time (OAT) method), one parameter is altered at a time, keeping all other parameters as base values. Due to high computational efficiency and simplicity, OAT methods have been extensively utilized for sensitivity analysis of hydrological models [9–12].

DRAINMOD is the most commonly used process-based model to simulate tile drainage systems, subsurface hydrology, and nitrogen and phosphorus dynamics in agricultural fields [2,13–16]. A few studies were conducted on the uncertainty analysis of hydrological and water quality parameters using DRAINMOD. For instance, Ref [17] made a relative sensitivity analysis of the DRAINMOD hydrological parameters in the study on tile drain spacing optimization in four fields in the Little Vermilion watershed, Illinois. Ref [9] performed a two-step sensitivity analysis using global variables to assess the sensitivity of nitrate losses on drainage using model prediction. Similarly, recently, Ref [15] made a sensitivity analysis for DRAINMOD-H and DRAINMOD-N modules using the Morris screening. The degree of the sensitivity of these parameters, however, differed. For example, Ref [17] considered maximum surface storage an insensitive parameter, but Ref [15] considered it a sensitive parameter for drainage flow. The possible reason for the differing results may be the variation in field conditions and soil types which the authors still need to explain in their study.

The researchers suggested that model calibration and uncertainty analysis based purely on one variable does not ensure the reliability of hydrological models since water balance elements might need to be more accurately represented [18]. The multi-objective calibration and sensitivity analysis of the hydrological models may reduce the uncertainty and help with the issue of equifinality in the model calibration process [19]. However, a few studies were performed regarding the multi-objective sensitivity analysis of DRAINMOD. So, this study aimed to enhance the model calibration and reduce the equifinality and parameter uncertainty of DRAINMOD using multi-objective sensitivity analysis of 17 hydrological parameters. The authors analyzed the local sensitivity of the DRAINMOD hydrological parameters for multiple objective functions representing drainage flow, water balance, and relative yield. Moreover, a comparative sensitivity evaluation was performed for two fields with distinct drainage designs, soil types, and other field conditions.

## 2. Materials and Methods

### 2.1. Study Area

The study was conducted at the South Farm Agricultural Research Station of the University of Illinois located at Champaign County in Illinois, USA (−88.211472 E, 40.053167 N). The 30-acre field was divided into six sub-fields as CF-1, CF-2, CF-3, CF-4, CF-5, and CF-6, and respective outlets as CS-1, CS-2, CS-3, CS-4, CS-5, and CS-6 with different drainage configurations to examine the impacts of tile drainage configurations on the drainage water, soil properties, and nutrient losses, as well as crop production (Figure 1). Instrumentation for the field was set up in May 2018, and the field monitoring of tile flow, nutrient concentration, and soil sampling was conducted for 2018–2022 to understand better the effects of tile spacing and depth on hydrological responses and crop production. CF-6 was located at the highest elevation (707 ft), while CF-1 was at the lowest elevation (692 ft). As per the United States Department of Agriculture Natural Resources Conservation Service (USDA-NRCS) soil survey report, the predominant soil for the field was Flanagan (154A), followed by Drummer (152A). This study considered two fields, CF-3 and CF-4, having distinct soil types: Drummer and Flanagan, respectively, with surface slopes being 1.2% for CF-3 and about 1.8% for CF-4. Rainfall, temperature, and subsurface flow measured at the study site were the observed data used for the model setup. The site was planted with corn for 2019, 2020, and 2022 and soybean for 2021 with conventional drainage practices.

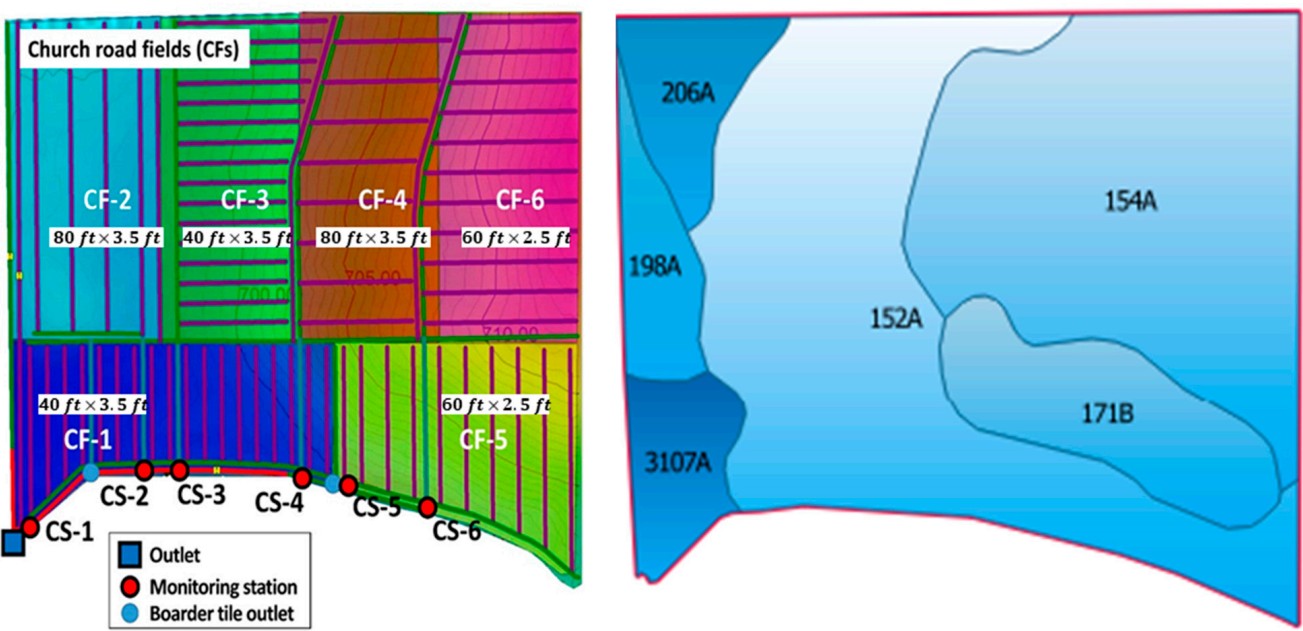

**Figure 1.** Study area showing the field partition, monitoring outlets, and soil classification.

### 2.2. Model and Parameters Description

DRAINMOD is a deterministic field-scale model to simulate hydrology, nitrogen, carbon, and phosphorus dynamics in poorly drained soils. The model was scaled up to the watershed scale by incorporating the surface runoff routes from the field and projecting the flow rates and stages in the drain channels and receiving streams [2,20]. DRAINMOD-H is the module of DRAINMOD for hydrological analysis that performs water balance on hourly and daily time scales, and the hydrologic variables such as surface runoff, infiltration, subsurface drainage, water table depth, and drained pore space in the soil profile can be projected on yearly, monthly, or daily scales as per the need. In addition, relative crop yield and irrigation water requirement can be predicted [21]. The fundamental relationship of the

model is the water balance in the unit area of soil layers that extend from the surface to the impermeable layer (Figure 2). For the time step of Δt, the water balance can be equated as

$$\Delta Va = F - ET - D - DS$$

and

$$P = F + RO + \Delta S$$

where,

ΔVa = Soil storage or change in the air volume (cm);
F = Infiltration (cm);
ET = Evapotranspiration (cm);
D = Lateral drainage (cm);
DS = Deep seepage (cm);
P = Precipitation (cm);
RO = Surface runoff (cm);
ΔS = Change in surface water storage (cm).

Characterization of infiltration was conducted using Green–Ampt equation [22] and potential evapotranspiration (PRT) was determined using Thornthwaite method [23] using temperature as the sole climatic data. Similarly, subsurface drainage flow (cm/h) was computed using Hooghoudt's steady state equation [24] with correction for the convergence near drains given by

$$q = \frac{4\,Ke \times m \times (2de + m)}{L^2}$$

where $q$ is the lateral subsurface drainage flux (cm/h), $Ke$ is the effective lateral hydraulic conductivity (cm/h) under water table, $m$ represents the midpoint of the height in water table above the drain (cm), $de$ is the equivalent depth which is the depth of impermeable layer below the base of drain (cm), and $L$ is the spacing between drains (cm).

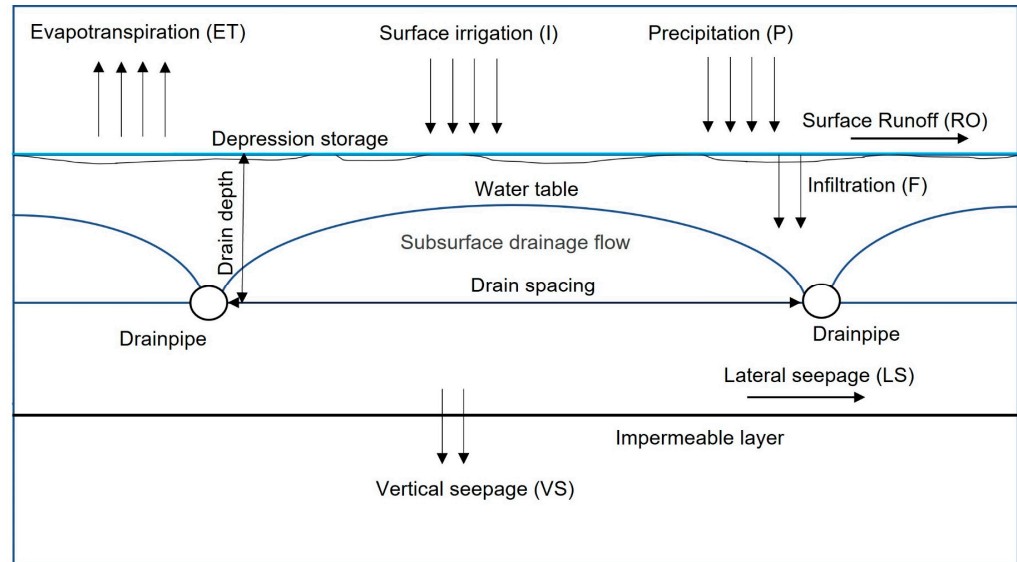

**Figure 2.** Schematic representation of sub-surface hydrological components used for DRAINMOD (Source: [25]).

Maximum surface storage (SS) characterizes the intensity of surface drainage which indicates the average depth of depression storage that needs to be filled before surface runoff takes place.

### 2.3. Data and Sources of Information

Daily climate data (temperature and rainfall) were measured in a nearby field with a weather station. Evapotranspiration was computed using the Thornthwaite equation using observed temperature and rainfall data [23]. Out of 17 hydrologic parameters considered in the study, some drainage system parameters, such as drain depth (B) and drain spacing (L), were obtained from the observed data. The drainage coefficient (DC) computed using the slope and drain area was less than the observed drainage flow. Hence, the maximum observed historical daily subsurface drainage was used as DC in the study. Soil layer profile and respective bottom layer depth were used as that of Drummer, a silty clay loam, from the United States Department of Agriculture (USDA), Natural Resources Conservation Services (NRCS) (Table 1). Corrugated plastic tile drains of diameter 4 inches were installed at 40 ft (12.2 m) spacing at a depth of 3.5 ft (1.07 m) at a slope of 1.2% for CS-3 and 80 ft spacing at a depth of 3.5 ft at the slope of 1.8% for CS-4. The daily drainage flow from the tiles was obtained from the continuously monitored data. Grain corn was planted in both fields on 24 April 2019, harvested on 7 October 2019, and planted on 8 April 2020 and harvested on 7 October 2020, while soybean was planted on 10 April 2021 and harvested on 10 October 2021. All the other coefficients and parameters for evapotranspiration, soil, and crops were initialized based on the past study in the nearby locations [14], which were later calibrated for the observed data.

**Table 1.** Parameters for DRAINMOD-H with their range and base parameter values for sensitivity analysis.

| Parameters | Meaning | Unit | Parameter Value | |
|---|---|---|---|---|
| **Drainage System Parameters** | | | CS3 | CS4 |
| H | Depth to Impermeable Layer | cm | 152 | 152 |
| Re * | Effective Radius | cm | 1.1 | 1.1 |
| B | Drain Depth | cm | 107 | 107 |
| L | Drain Spacing | cm | 1220 | 2438 |
| SS * | Maximum surface storage | cm | 1.2 | 0.9 |
| KD * | Kirkham Depth | cm | 1.76 | 2.2 |
| DC | Drainage coefficient | cm/day | 2.09 | 2.36 |
| **Soil properties** | | | | |
| | Lateral saturated conductivity | | | |
| LK5 * | layer 5 | cm/h | 0.1 | 0.69 |
| LK4 * | layer 4 | cm/h | 3 | 1.5 |
| LK3 * | layer 3 | cm/h | 5.5 | 0.26 |
| LK2 * | layer 2 | cm/h | 2 | 1.09 |
| LK1 * | layer 1 | cm/h | 3 | 1.88 |
| | Soil type | | Drummer | Flanagan |
| | Soil layer bottom depth | | | |
| | layer 5 | cm | 152 | 152 |
| | layer 4 | cm | 100 | 114 |
| | layer 3 | cm | 81 | 97 |
| | layer 2 | cm | 48 | 58 |
| | layer 1 | cm | 18 | 46 |
| Slope | Surface slope | % | 1.2 | 1.8 |
| | Surface length along drain tiles | cm | 7800 | 8600 |
| **Soil temperature parameters** | | | | |
| ZA * | ZA coefficient | | 3.9 | 7.64 |
| ZB * | ZB coefficient | | 1.4 | 1.4 |
| TKA * | Thermal conductivity function (TKA) | | 3 | 3.97 |
| TKB * | Thermal conductivity function (TKB) | | 1.3 | 0.26 |
| T_dep | Soil temperature at bottom of soil profile | °C | 11.5 | 11.5 |
| T_snow | Avg air temp below which precipitation is snow | °C | 0 | 0 |
| T_melt | Average air temp above which snow starts to melt | °C | 1 | 1 |
| CDEG * | Snow melt coefficient | mm/dd-°C | 7.6 | 3.58 |
| CICE * | Critical ice content above which infiltration stops | $cm^3/cm^3$ | 0.2 | 0.11 |

**Table 1.** *Cont.*

| Parameters     Meaning | | Unit | | Parameter Value | |
| --- | --- | --- | --- | --- | --- |
| **Drainage System Parameters** | | | | **CS3** | **CS4** |
| Objective functions | | CS-3 | | CS-4 | |
| | | Calibration | Validation | Calibration | Validation |
| NSE | | 0.5 | 0.58 | 0.49 | 0.46 |
| RSQ | | 0.5 | 0.645 | 0.53 | 0.52 |
| RMSE | | 0.206 | 0.161 | 0.196 | 0.192 |
| PBIAS | | −5.70% | −28% | 1.86% | −25% |

* Represents calibrated parameters.

### 2.4. Calibration and Validation

The DRAINMOD model was run from January 2017 to December 2022 with a warmup period from January 2017 to November 2018. The calibration of DRAINMOD parameters was conducted for daily and annual subsurface flow for the period of November 2018 to December 2020 for continuous corn production, and the validation was carried out for the period of January 2021 to May 2022. The model was initialized using known data as well as available literature for unknown parameters. First, a range of parameters was selected based on the literature and judgment based on the field conditions (Table 1). The flow parameters were divided into an array for the step interval of approximately 5%. The Monte Carlo simulation was performed by running DRAINMOD 2000 times each to obtain the best set of parameters using integrated Python codes and DRAINMOD. The parameter range was updated by sensitivity analysis and graphical plots of the Monte Carlo simulation results. The process was repeated till the model outputs well represented the observed flow. Some manual adjustments were made for the final calibration of the model on the lateral hydraulic conductivity and maximum surface storage following both the graphical and statistical approaches.

Absolute values of Nash–Sutcliffe model efficiency coefficient (NSE), coefficient of determination (RSQ), root mean square efficiency (RMSE), and percentage bias (PBIAS) for the daily subsurface drainage flow and total cumulative flow were the objective functions considered for calibration of the model. RSQ ranges from 0 to 1 and indicates the correlation between observed and simulated data series of daily drainage flow, with 1 being the best correlation [26]. NSE ranges from $-\infty$ to 1 and represents the closeness between the data series [27]. RMSE quantifies the average magnitude of differences between simulated and observed values, while PBIAS denotes whether the simulated outputs are overestimated or underestimated [28]. Calibration was considered satisfactory when the objective functions for NSE and RSQ were greater than 0.4 for the daily hydrological data, and the difference between cumulative flow was less than 10%. The graphical plot of daily hydrology data and cumulative flow was used to assess the anomalies between the simulated and observed values, for instance, in peak flow and slope of cumulative flow data.

$$NSE = 1 - \frac{\sum_{i=1}^{n}(O_i - S_i)^2}{\sum_{i=1}^{N}(O_i - \overline{O})^2}$$

$$R^2 = \frac{\sum_{i=1}^{n}(O_i - \overline{O}) * (S_i - \overline{S})}{\sqrt{\sum_{i=1}^{n}(O_i - \overline{O})^2} * \sqrt{\sum_{i=1}^{n}(S_i - \overline{S})^2}}$$

$$PBIAS = \frac{\sum_{i=1}^{n}(O_i - S_i)}{\sum_{i=1}^{n} O_i} \times 100$$

$$RMSE = \sqrt{\sum_{i=1}^{n}(O_i - S_i)^2 / n}$$

where $O_i$ is the observed value, $S_i$ is the simulated value and $\overline{O}$ is the mean of observed values, $\overline{S}$ is the mean simulated values and n is the number of data.

### 2.5. Sensitivity Analysis of Parameters

A total of 17 flow hydrological parameters, including drainage system parameters, soil hydraulic parameters, and soil temperature parameters, were selected for the sensitivity analysis using Latin Hypercube One-Factor-at--Time (LH-OAT) method, which is commonly utilized for screening and sensitivity analysis. In this method, only one parameter varies. In contrast, the other parameters remain constant, which allows a specific attribute to the change in model output concerning change in the input parameters. Sensitive analysis was performed for multiple objective functions: NSE, RSQ, and RSME representing daily drainage flow, PBIAS, and total cumulative flow representing long-term water balance and relative yield of corn. The relative yield of corn was taken as an average of 2019 and 2020 from the DRAINMOD output. Two types of sensitivity indicators were calculated: relative and absolute.

Relative sensitivity Index (RSI) defined by [9] was used for quantitative evaluation of the sensitivity of the model output with an absolute change in the model inputs.

$$Sij = \frac{|O(x1, \ldots xi + \Delta xi, \ldots xp) - O(x1, \ldots xi, \ldots xp)|}{(O(x1, \ldots xi + \Delta xi, \ldots xp) + O(x1, \ldots xi, \ldots xp))/2} * \frac{xi}{|\Delta xi|}$$

Absolute sensitivity Index was defined as

$$Sij = \frac{|O(x1, \ldots xi + \Delta xi, \ldots xp) - O(x1, \ldots xi, \ldots xp)|}{|\Delta xi|}$$

where $S_{ij}$ is the relative partial effect of a parameter ($x_i$) out of total $p$ parameters and O refers to the output of model for the considered objective functions. Partial sensitivity indices (both relative and absolute) were considered in terms of magnitudes only rather than considering signs for eliminating the cumulative effects.

The parameter $x_i$ was then varied by factor $\Delta x_i = 5\%$ from the range $-80\%$ to $+80\%$ of the base value in the step interval of 5%. The best parameter sets after calibration were taken as the base value, and the simulation was repeated. The final sensitivity index $S_{xi}$ was then computed by taking the average of these partial effects. The higher the value of $S_{xi}$ represents the outputs are more sensitive to the given parameters.

$$Sxi = \frac{\sum_{j=1}^{N} Sij}{N}$$

The sensitivity analysis was performed for both fields by repeating the same procedure.

## 3. Results and Discussion

### 3.1. Calibration and Model Performance

The known parameters and calibrated results are presented in Table 1. Overall, the calibration was good for both CS-3 and CS-4 for November 2018 to December 2020. The NSE and RSQ for daily flow for CS-3 were 0.50 and 0.50, respectively, while that for field CS-4 were 0.49 and 0.53, respectively. Similarly, the difference in observed and simulated cumulative flow for field CS-3 was $-5.70\%$, and that of field CS-4 was 1.86%. The simulated cumulative flow was underpredicted (PBIAS = $-5.70\%$ for calibration and PBIAS = $-28\%$ for validation) for CS-3, while the flow was slightly overpredicted (PBIAS = 1.86% for calibration and $-25\%$ for validation) for CS-4. High PBIAS values for these fields could be because of low total outflow and inefficient simulation of peaks at the validation period. The calibrated model was validated from January 2021 to December 2021. NSE and RSQ for field CS-3 for validation were 0.58 and 0.62, respectively, that of field CS-4 was 0.46 and 0.52, respectively. Though the peaks were not well-simulated, water balance, correlation,

and closeness were good enough to represent the observed data for the overall study period (Figure 3).

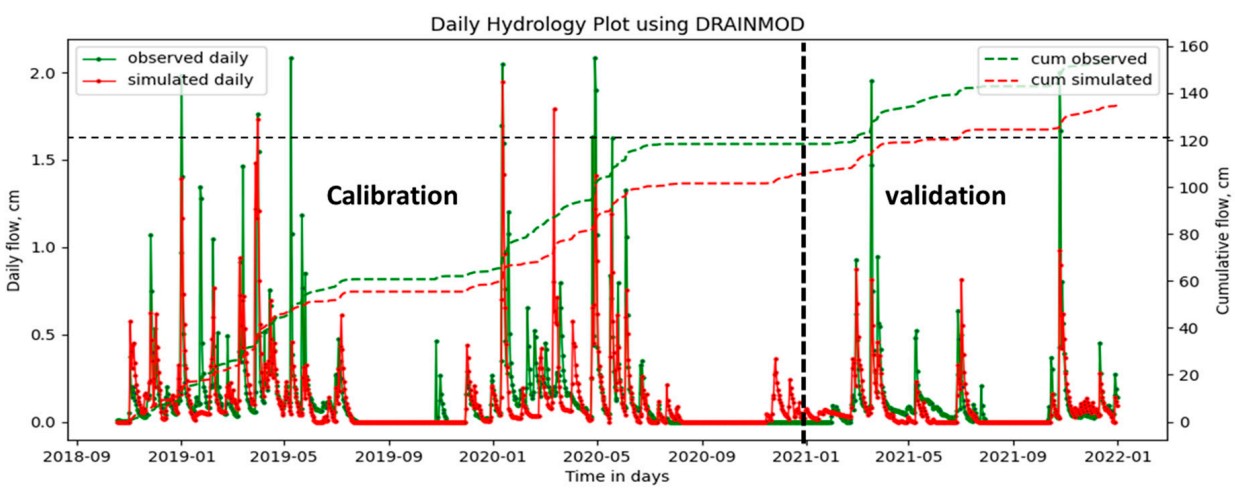

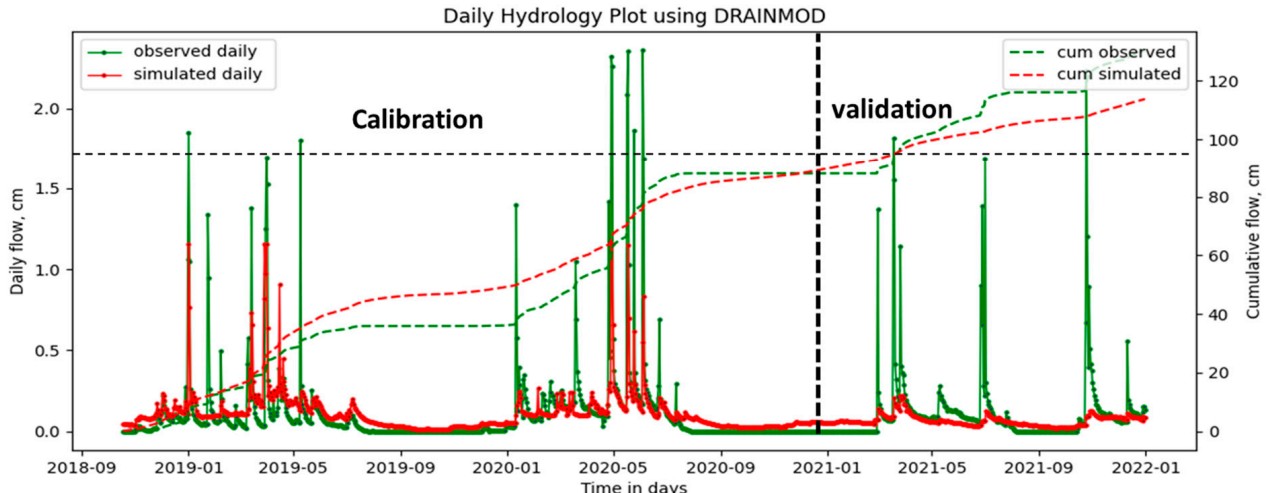

**Figure 3.** Observed and simulated daily drainage flow for CS-3 (**upper**), CS-4 (**lower**) for study period.

### 3.2. Sensitivity Analysis

The scatter plot for the DRAINMOD output with the change in input parameters for CS-3 suggests that the daily flow was most sensitive to the drainage design parameters such as drain spacing and depth. The parametric variation from −80% to 0 had much steeper slopes (Figure 4) than the variation from 0 to 80%. This finding indicates that the daily flow pattern, both in terms of closeness (NSE) as low as 0.4 and correlation (RSQ) as low as 0.6, are much different at the lower drain spacing and drain depth. However, the close drain spacing than the field settings (1220 cm) did not impact the overall cumulative flow much, as shown by the cumulative flow and PBIAS plot. A possible reason could be the sufficiency of the drainage system to remove the infiltrated water at this design.

On the other hand, drain depth depicted remarkable effects in both daily flow and cumulative water balance, as well as the relative yield of corn. A decrease in drain depth by 80% yielded 30% less flow, and relative yield decreased to 20%. The yield is impacted by drought or excessive water conditions, salinity, and planting delay. However, in this study, the effects of salinity and planting delay were not evaluated. The reduction in the relative yield is solely attributed to the excessive water stress on the crop due to ineffective removal at the root zone. Drainage coefficient (DC), a parameter indicating the capacity of drainage removal, impacted the flow to a specific range only. For instance, in CS-3 (Figure 4), figures for NSE and RSQ had a sharp drop from −50% to −80% due to the tile drains' limiting

hydraulic capacity to remove the maximum infiltration volume to transport away from the field. Kirkham depth impacted the cumulative flow results even if the daily flow indicators had the same output, while surface storage and slope did not vary the model outputs significantly at the given range.

**DRAINMOD output sensitivity with parametric variation for CS−3**

**Figure 4.** DRAINMOD output sensitivity with parametric variation for field CS-3.

Soil parameters such as lateral conductivity of two different soil types had varying effects on both flow and relative yield. The lateral hydraulic conductivity of the third layer (LK3) from the surface impacted outputs of CS-3, while that of other layers was insignificant. However, for CS-4 (Figure 5), LK3 and LK2 were insignificant, while LK5, LK4, and LK1 held viable impacts on the flow and yield outputs. This was mainly due to the varying thickness of each layer for Drummer and Flanagan soil types. For instance, in Drummer soil, the thickness of the first layer was 18 cm while that in Flanagan was 46 cm. More variability in output was observed due to the change in the lateral conductivity of layer 1 in Flanagan. Layer 4, which was the layer consisting of tile drains, and layer 5, which was a layer just below the tile drains, had a notable effect on the model output of CS-4. For Drummer soil, layer 5 was the thickest layer and consisted of tile drains. However, it did not impact the local sensitivity of outputs. However, it played a significant role during model calibration using Monte Carlo simulation. This phenomenon is shown due to the non-additive and non-linearity of the model, whose sensitivity is only realized from global sensitivity analysis using the interaction of multiple parameters at once.

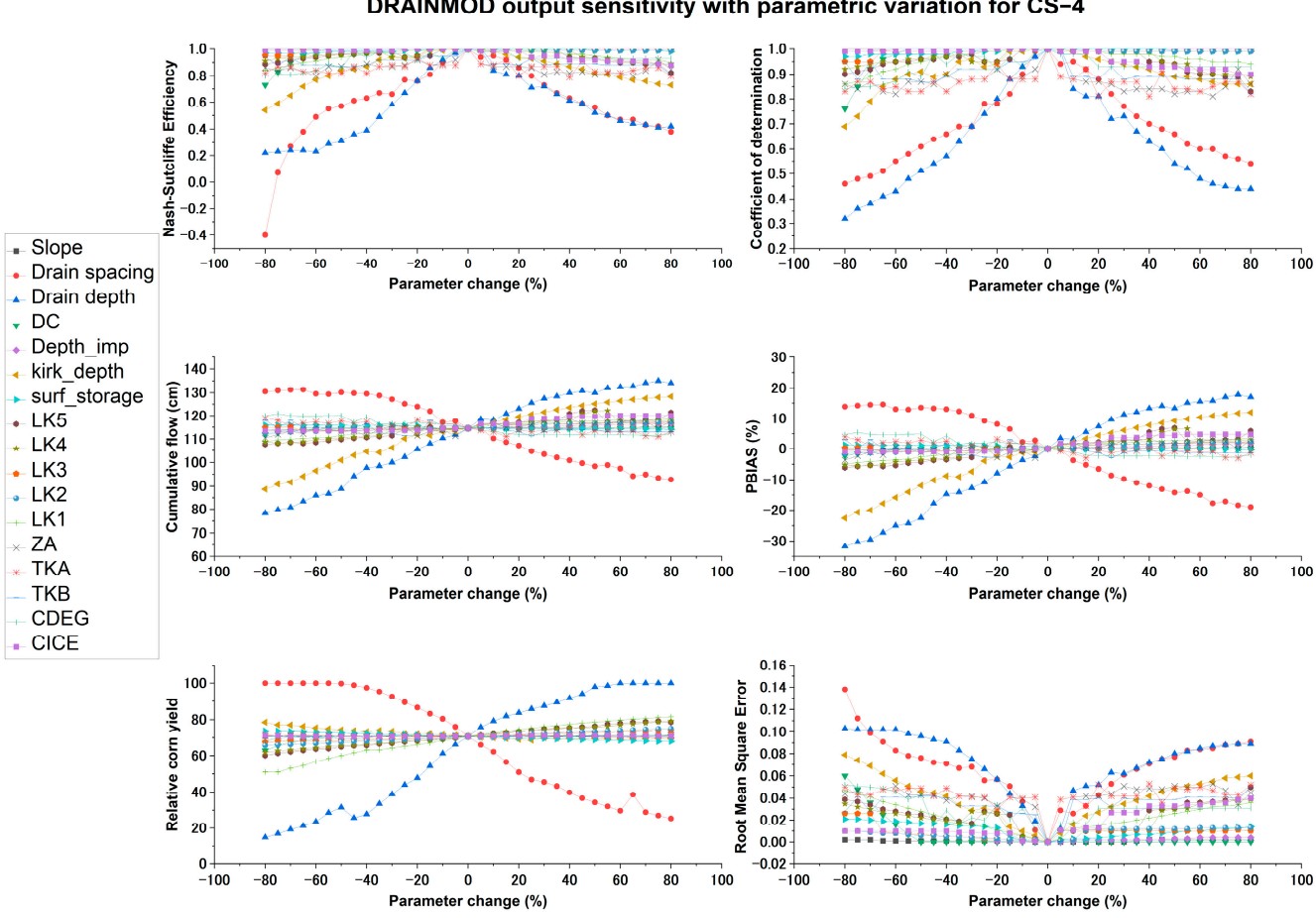

**Figure 5.** DRAINMOD output sensitivity with parametric variation for field CS-4.

Soil temperature parameters such as CICE, CDEG, and ZA did not impact the relative yield but significantly affected the flow outputs. The effects were more notable on the water balance as represented by a steep gradient for cumulative flow (Figures 4 and 5). CICE is associated with the critical ice content above, which infiltration stops, and CDEG with the rate of ice melt, which plays a role in water balance in the soil layers during winters. Thus, these parameters do not contribute to the relative yield of corn. However, their uncertainty impacts winter drainage flow and, so, their impact on water quality might be significant. TKA and TKB represent soil thermal conductivity function and had little impact on both model outputs.

Figures 6 and 7 summarize the LH-OAT sensitivity index value of DRAINMOD hydrologic parameters for CS-3 and CS-4, respectively. RMSE and PBIAS are represented by the absolute sensitivity index, while the relative sensitivity index characterizes NSE, RSQ relative yield, and cumulative flow. It is because the values of RMSE and PBIAS are zero at base simulation, which cannot represent the actual relative sensitivity of each parameter since the ratio of change in model output per change in parameters input would always be constant, whatever the parameters be. In both figures, the sensitivity of PBIAS appears more dominating than RMSE. The dominating magnitude is because of the higher absolute values of PBIAS than RMSE. However, the relative sensitivity characteristics can depict the comparative evaluation of the model output's behavior. Most RSI values were higher for daily flow indicators (NSE and RSQ) than the water balance indicator (cumulative flow) in both fields. It represents the lower agreement of the daily flow between the base and simulated values than the long-term water balance by varying parameters in each model run.

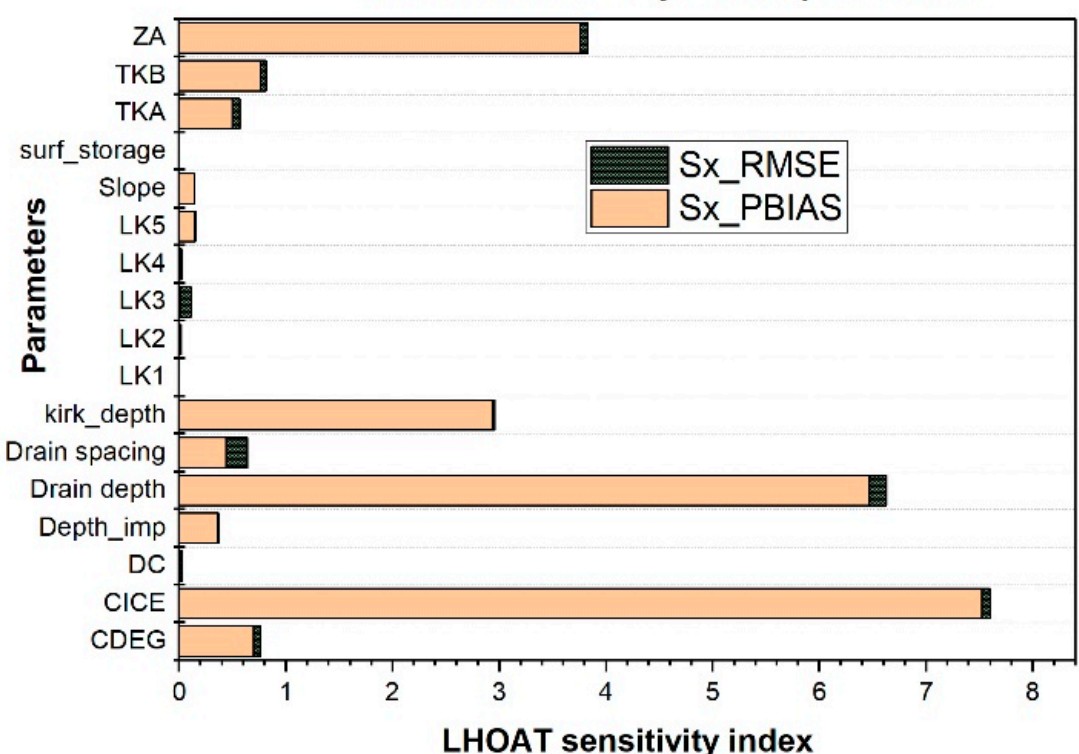

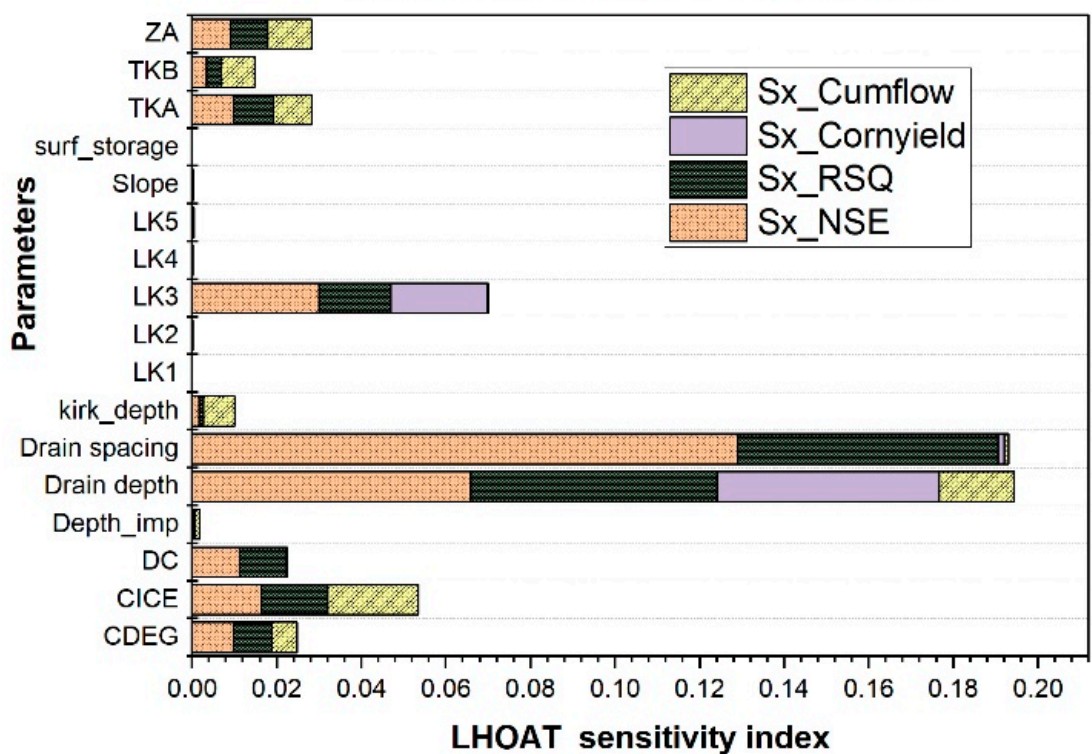

**Figure 6.** LH-OAT sensitivity index for CS-3.

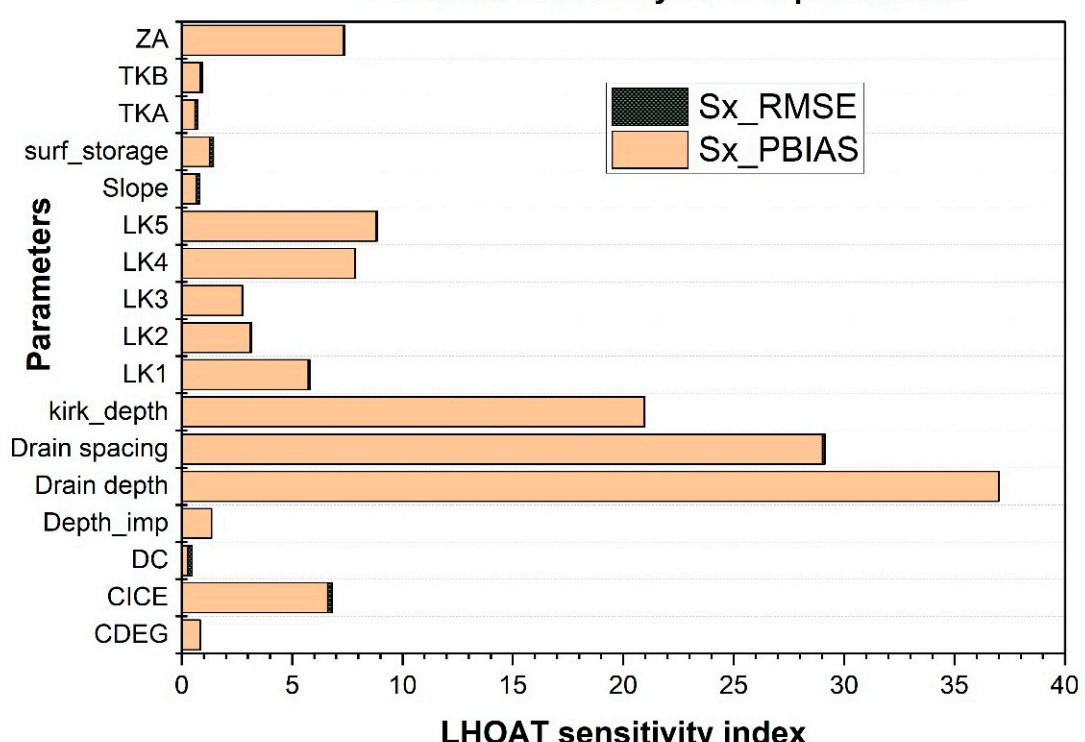

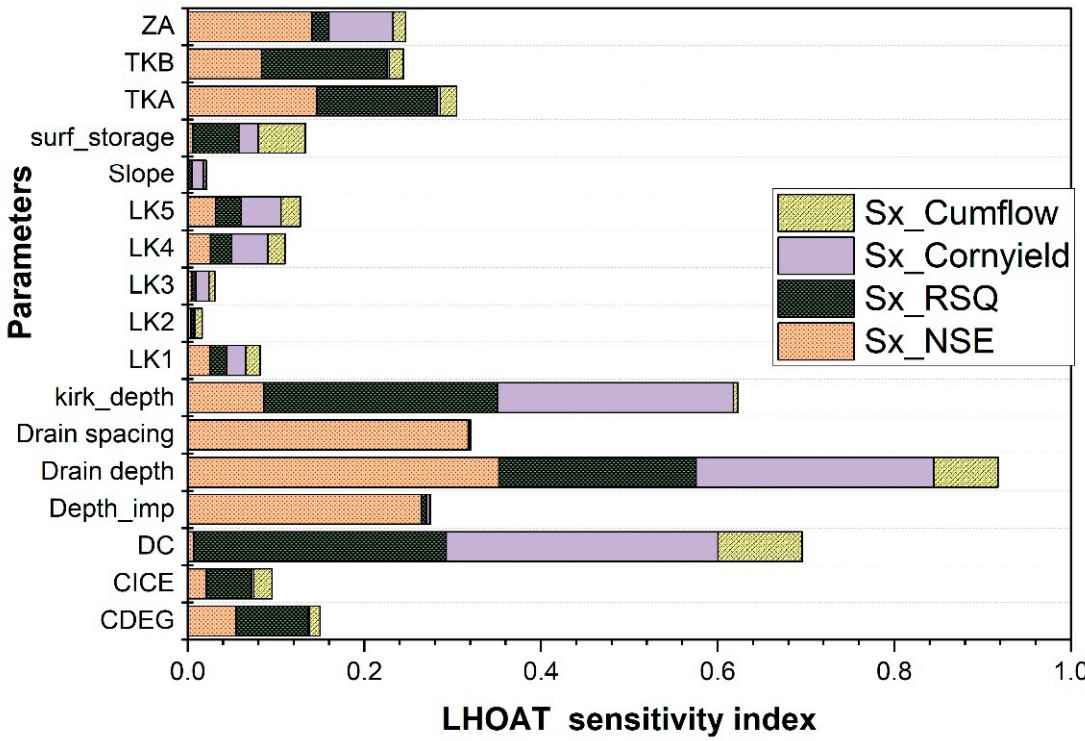

**Figure 7.** LH-OAT sensitivity index for CS-4.

For the CS-3 field, daily flow was most sensitive to drain spacing (ranked first for NSE and RSQ sensitivity), followed by drain depth and lateral hydraulic conductivity of the third layer from the surface (Table 2). Similarly, LH-OAT ranked CICE and DC as the fourth and fifth sensitive parameters for daily flow. Nevertheless, RSQ and NSE did not behave similarly in field CS-4 (Table 3). RSI for NSE was highest for drain depth, followed by drain spacing, depth of impermeable layer, TKA, ZA, and Kirkham depth, respectively. However, drain spacing ranked first, followed by the depth of impermeable layer, drain spacing, TKA, and ZA. Slope and surface storage had negligible effects on the model outputs in both fields. In general, the slope affects the runoff and, hence, the infiltration capacity of the field, but the slopes in both fields were less than 2%, which is comparatively flat, and the length of fields was also short (7.8 m for CS-3 and 8.6 m for CS-4) in the direction of drain tiles layout. Thus, the variation of slopes within the set ranges did not affect the model outputs significantly. The lateral conductivity of the second and third layers also had an insignificant contribution to flow outputs for CS-4. In CS-4, these were layers with comparatively lower thicknesses than the other layers. Moreover, the soil layers' composition (clay, silt, and sand percentage) and water-holding capacity also impacted the layer properties.

**Table 2.** Compiled LH-OAT sensitivity ranking for CS-3.

| Parameters | Relative Sensitivity Index | | | | | | | | Absolute Sensitivity Index | | | |
|---|---|---|---|---|---|---|---|---|---|---|---|---|
| | NSE | | RSQ | | Yield | | Cum flow | | PBIAS | | RMSE | |
| | Sx | Rank | Sx | Rank | Sx | Rank | Sx | Rank | Sx | Rank | Sx | Rank |
| CDEG | 0.010 | 6 | 0.009 | 7 | 0.000 | 16 | 0.006 | 7 | 0.696 | 6 | 0.064 | 7 |
| CICE | 0.016 | 4 | 0.016 | 4 | 0.000 | 17 | 0.021 | 1 | 7.523 | 1 | 0.073 | 5 |
| DC | 0.011 | 5 | 0.011 | 5 | 0.000 | 4 | 0.000 | 15 | 0.001 | 14 | 0.024 | 9 |
| Depth_imp | 0.000 | 11 | 0.000 | 11 | 0.000 | 6 | 0.001 | 8 | 0.363 | 9 | 0.002 | 14 |
| Drain depth | 0.066 | 2 | 0.058 | 2 | 0.053 | 1 | 0.018 | 2 | 6.466 | 2 | 0.158 | 2 |
| Drain spacing | 0.129 | 1 | 0.062 | 1 | 0.001 | 3 | 0.001 | 9 | 0.437 | 8 | 0.197 | 1 |
| Kirk_depth | 0.002 | 10 | 0.001 | 10 | 0.000 | 7 | 0.007 | 6 | 2.933 | 4 | 0.023 | 10 |
| LK1 | 0.000 | 17 | 0.000 | 17 | 0.000 | 12 | 0.000 | 17 | 0.000 | 16 | 0.000 | 16 |
| LK2 | 0.000 | 12 | 0.000 | 16 | 0.000 | 11 | 0.000 | 14 | 0.000 | 15 | 0.016 | 11 |
| LK3 | 0.030 | 3 | 0.017 | 3 | 0.023 | 2 | 0.000 | 12 | 0.012 | 13 | 0.099 | 3 |
| LK4 | 0.000 | 16 | 0.000 | 15 | 0.000 | 10 | 0.000 | 13 | 0.012 | 12 | 0.009 | 12 |
| LK5 | 0.000 | 15 | 0.000 | 14 | 0.000 | 9 | 0.000 | 10 | 0.145 | 10 | 0.007 | 13 |
| Slope | 0.000 | 13 | 0.000 | 12 | 0.000 | 5 | 0.000 | 11 | 0.140 | 11 | 0.000 | 15 |
| Surf_storage | 0.000 | 14 | 0.000 | 13 | 0.000 | 8 | 0.000 | 16 | 0.000 | 17 | | 17 |
| TKA | 0.010 | 7 | 0.009 | 6 | 0.000 | 14 | 0.009 | 4 | 0.495 | 7 | 0.073 | 6 |
| TKB | 0.003 | 9 | 0.003 | 9 | 0.000 | 15 | 0.008 | 5 | 0.758 | 5 | 0.055 | 8 |
| ZA | 0.009 | 8 | 0.009 | 8 | 0.000 | 13 | 0.010 | 3 | 3.756 | 3 | 0.074 | 4 |

For the CS-3 field, the long-term water balance was most sensitive to soil temperature parameters with the first ranking of CICE (Table 2). Similarly, other soil temperature parameters ZA, TKA, and TKB were ranked third, fourth, and fifth for CS-3. This result was quite unusual compared to other flow indices and the water balance index for field CS-4. A possible reason could be the winter flow observed from the model output compared to the no-flow condition from the observed data (Figure 3). This result also depicts an uncertainty of soil temperature parameters affecting the behavior of sub-surface hydrology in cold regions such as Illinois. For CS-4, however, water balance was most affected by the drainage system parameters drain depth and spacing with first and second ranking, respectively. This result corresponds to the daily flow indicator (NSE) for CS-4. Kirkham depth, LK5, and CDEG were ranked third, fourth, and fifth, respectively. The possible reason for discrepancies in the water balance indicators for CS-3 and CS-4 could be soil composition and variation in the thickness of the soil layers.

**Table 3.** Compiled LH-OAT sensitivity ranking for CS-4.

| Parameters | Relative Sensitivity | | | | | | | | Absolute Sensitivity | | | |
| | NSE | | RSQ | | Yield | | Cum flow | | PBIAS | | RMSE | |
| | Sx | Rank | Sx | Rank | Sx | Rank | Sx | Rank | Sx | Rank | Sx | Rank |
|---|---|---|---|---|---|---|---|---|---|---|---|---|
| CDEG | 0.007 | 13 | 0.006 | 13 | 0.003 | 12 | 0.001 | 17 | 0.829 | 14 | 0.101 | 7 |
| CICE | 0.000 | 17 | 0.000 | 17 | 0.001 | 17 | 0.002 | 16 | 6.602 | 7 | 0.052 | 11 |
| DC | 0.006 | 14 | 0.005 | 14 | 0.013 | 10 | 0.003 | 15 | 0.267 | 17 | 0.008 | 15 |
| Depth_imp | 0.264 | 3 | 0.264 | 2 | 0.267 | 3 | 0.005 | 14 | 1.350 | 11 | 0.005 | 16 |
| Drain depth | 0.005 | 15 | 0.005 | 15 | 0.015 | 9 | 0.007 | 13 | 37.005 | 1 | 0.213 | 1 |
| Drain spacing | 0.003 | 16 | 0.003 | 16 | 0.021 | 8 | 0.008 | 12 | 29.023 | 2 | 0.200 | 2 |
| kirk_depth | 0.084 | 7 | 0.081 | 6 | 0.002 | 15 | 0.012 | 11 | 20.942 | 3 | 0.107 | 6 |
| LK1 | 0.025 | 11 | 0.019 | 11 | 0.072 | 4 | 0.014 | 10 | 5.734 | 8 | 0.059 | 10 |
| LK2 | 0.146 | 4 | 0.142 | 4 | 0.003 | 14 | 0.016 | 9 | 3.106 | 9 | 0.023 | 14 |
| LK3 | 0.021 | 12 | 0.019 | 12 | 0.002 | 16 | 0.016 | 8 | 2.739 | 10 | 0.024 | 13 |
| LK4 | 0.141 | 5 | 0.136 | 5 | 0.004 | 11 | 0.018 | 7 | 7.845 | 5 | 0.061 | 9 |
| LK5 | 0.026 | 10 | 0.024 | 10 | 0.041 | 6 | 0.020 | 6 | 8.789 | 4 | 0.068 | 8 |
| Slope | 0.055 | 8 | 0.051 | 8 | 0.003 | 13 | 0.021 | 5 | 0.650 | 15 | 0.003 | 17 |
| surf_storage | 0.031 | 9 | 0.029 | 9 | 0.045 | 5 | 0.022 | 4 | 1.259 | 12 | 0.029 | 12 |
| TKA | 0.086 | 6 | 0.052 | 7 | 0.022 | 7 | 0.053 | 3 | 0.598 | 16 | 0.176 | 3 |
| TKB | 0.317 | 2 | 0.223 | 3 | 0.269 | 2 | 0.073 | 2 | 0.837 | 13 | 0.123 | 5 |
| ZA | 0.352 | 1 | 0.286 | 1 | 0.307 | 1 | 0.096 | 1 | 7.332 | 6 | 0.172 | 4 |

Field CS-3 had a better drainage design than field CS-4 (Table 1), and the base relative yield was close to 100%. Thus, given the ranges of parameters, other parameters except the drainage design parameter did not significantly impact the relative yield of corn. Though the yield output of corn for 2019 and 2020 differed, the average yield was taken as an indicator of overall corn yield. For CS-3, drain depth, LK3, and drain spacing impacted the corn yield. However, CS-4 needed a better drainage design, so the relative yield could have been higher (average 70%). Also, many other parameters, in addition to drainage design, impacted the yield of corn for this design. Like CS-3, drain depth was the most sensitive parameter influencing relative yield, followed by drain spacing. These are visible in the relative yield plot (Figure 5), where the increase in drain depth and decrease in drain spacing improved the relative yield by up to 100%. Relative yield for CS-4 was also sensitive to the depth of the impermeable layer and lateral hydraulic conductivities of the first, fifth, and fourth layers. Given the inadequate drainage design, lateral conductivities of these layers affected the water logging in these soil layers at the root zone. In both fields, soil temperature parameters were not sensitive to corn's relative yield.

### 4. Conclusions and Limitations

Calibration, validation, and sensitivity analysis of 17 hydrologic parameters for DRAINMOD were performed in two fields with distinct soil characteristics, and drainage design was conducted. The multi-objective sensitivity analysis of the fields using daily drainage flow, water balance, and relative yield depicts the varying results for these fields. The results indicated that both daily and long-term flow, as well as the relative yield of the corn, were most sensitive to the drainage design parameters of the tile drain spacing and drain depth. Parameters related to the soil properties influencing sub-surface hydrology, such as the lateral hydraulic conductivity of dominant layers, impacted the flow and yield results. Soil temperature-related parameters mainly impacted the long-term water balance but did not affect the relative yield of corn. These findings, however, may differ in other regions since this study was conducted in specific climatic regions.

Since sensitivity analysis reflects the parameter uncertainty of the model, careful calibration of the most sensitive parameters needs to be conducted to reduce prediction errors. The study also suggests that drainage design variables such as depth and spacing must be considered cautiously depending on field settings, as they highly impact flow and

crop productivity. Some parameters were affected during the calibration of the model in combination with other parameters but not in the LH-OAT analysis, whose actual impacts might be realized by global sensitivity analysis. Moreover, we did not consider the nutrient parameters that could influence the relative crop yield, which we acknowledge was a limitation in the current study.

**Author Contributions:** Conceptualization, H.T., S.H. and R.B.; methodology, S.H.; software, H.T. and S.H.; validation, S.H. and R.B.; formal analysis, H.T.; resources, R.A.C. and R.B.; data curation, S.H.; writing—original draft preparation, H.T.; writing—review and editing, S.H. and R.B.; visualization, H.T.; supervision, S.H.; project administration, R.A.C. and R.B.; funding acquisition, R.A.C. and R.B. All authors have read and agreed to the published version of the manuscript.

**Funding:** This study was supported by the Illinois Nutrient Research & Education Council (project # 2018-3-360624-356). A partial support was provided by the National Institute of Food and Agriculture, U.S. Department of Agriculture, Hatch project (No. ILLU-741-337).

**Institutional Review Board Statement:** Not applicable.

**Informed Consent Statement:** Not applicable.

**Data Availability Statement:** Data are available upon request from the corresponding author.

**Acknowledgments:** Gratitude is extended to the University of Illinois at Urbana Champaign, Department of Agricultural and Biological Engineering. We gratefully acknowledge the comments and suggestions of the anonymous reviewers.

**Conflicts of Interest:** The authors declare no conflict of interest.

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
