# Peer review of "Comparative Sensitivity Analysis of Hydrology and Relative Corn Yield under Different Subsurface Drainage Design Using DRAINMOD"

_applsci, doi:10.3390/app13169252_

Round 1

Reviewer 1 Report

This is an interesting study. The paper is generally well written and structured. However, in my opinion the paper will be accepted after minor revision.

-I suggested to minimize the introduction.

-Make only the important results in abstract and reworded the abstract

-Minimize conclusions with the important findings 

Author Response

The authors would like to thank the reviewers for their valuable comments and suggestions. The comments are responded to and updated to the manuscript. The comments helped in escalating the contents of the paper and hence strengthen its quality.

Reviewer 2 Report

Reviewer Comments

Manuscript:

 Comparative Sensitivity Analysis of Hydrology and Relative Corn Yield under Different Subsurface Drainage Design using DRAINMOD.

 This research provides valuable information about the analysis of Hydrology and Relative Corn Yield under Different Subsurface Drainage Design using DRAINMOD. Manuscript is well written.

Minor Comments

Authors should mention out of six field why considers only two fields (CF-3 and CF-4)

Minor Comments

Line 129: one variable does not ensure …..Correct sentence with…………. one variable do not ensure

Author Response

(The authors gave the same response as above.)

Reviewer 3 Report

The provided description of the study on the sensitivity analysis of DRAINMOD hydrologic parameters for two field settings in Champaign, Illinois, offers valuable insights into the model's behavior and its implications for drainage system design and crop productivity. However, there are a few critical comments to consider:

1. The description does not provide sufficient details about the methodology employed for the sensitivity analysis. It mentions the use of Latin Hypercube-One factor at a time (LH-OAT), but no further information is provided on how the analysis was conducted, the sample size used, or the specific steps followed. Without these details, it is challenging to assess the robustness and reliability of the sensitivity analysis.

2. The study focuses on six objective functions related to flow, water balance, and corn yield. While these objectives are relevant, other important aspects, such as nutrient transport, water quality, and ecological impacts, are not considered. To provide a comprehensive understanding of the model's performance and its implications for drainage system design, these additional factors should be taken into account.

3. The description does not mention whether the sensitivity analysis results were compared with field data or measurements. Without such comparisons, it is difficult to assess the accuracy and reliability of the model's predictions. Field validation is crucial to determine the model's ability to capture real-world conditions accurately.

4. The study focuses on two field settings in Champaign, Illinois, which may limit the generalizability of the findings to other regions or soil types. Hydrologic processes and the effectiveness of drainage systems can vary significantly depending on climatic conditions, topography, and soil characteristics. Therefore, caution should be exercised when extrapolating the results to other locations.

Addressing these critical comments by providing more detailed methodology, incorporating field validation, considering a broader range of objectives, discussing parameter calibration practices, and discussing the practical implications of the findings would strengthen the study's scientific rigor and relevance.

Minor editing of English language required

Author Response

(The authors gave the same response as above.)

Reviewer 4 Report

The article with the title “Comparative Sensitivity Analysis of Hydrology and Relative Corn Yield under Different Subsurface Drainage Design using DRAINMOD” is interesting, and the scope deals with the Applied Sciences journal particularly for the special issue. The authors analyzed the local sensitivity of the DRAINMOD hydrological parameters for multiple objective functions representing drainage flow, water balance, and relative yield. However, there are general issues that should be addressed:

-       The authors said they used six objective functions, i.e., NSE, RSQ, RMSE, PBIAS, total cumulative flow and relative yield. I think objective function number 1-4 are not equal to 5-6. The objective function number 1-4 are commonly used to compare the model and observed data. However, objective function number 5-6 are not commonly used. In addition, there are no equations clearly for the objective function number 5-6. The author should explain and address this issue as well as add the objective function number 5-6.

-       The reason used NSE, RQS, RMSE and PBIAS should be clearly stated. Why to much objective function are used, sometime RSQ and RMSE are enough.

-       The abstract is not presented well. The term of infiltration stops (CICE), ZA coefficient, and snow melt coefficient (CDEG) were not clearly defined before, and they are existing in the results part without explanation in the method part.

-       The evapotranspiration was computed by Thornthwaite equation. This method was not standard as adopted by the FAO. Why the authors used this model should be explained since the authors used weather station that commonly many sensors used such as relative humidity and solar radiation. So, the author can compute ET model by the standard of FAO, Penman-Monteith model.

-       The conclusion is too long. Conclusion is presented to address the objective. I think it is better to make shorter and add Discussion part.

R2 is RSQ is equal? It should be consistent with the symbol.

Author Response

(The authors gave the same response as above.)

Round 2

Reviewer 3 Report

The authors have made all the required changes. Hence the MS can be accepted.

Minor editing of English language required

Reviewer 4 Report

The authors have responded and improved the article, I think it can be accepted